

# Comparative sequence analysis elucidates the evolutionary patterns of *Yersinia pestis* in New Mexico over thirty-two years

Mary E. Warren[1], Brett E. Pickett[1], Byron J. Adams[2,3], Crystal Villalva[1], Alyssa Applegate[1] and Richard A. Robison[1]

[1] Department of Microbiology and Molecular Biology, Brigham Young University, Provo, Utah, United States
[2] Department of Biology, Brigham Young University, Provo, UT, United States
[3] Monte L. Bean Life Science Museum, Provo, UT, United States

## ABSTRACT

**Background:** *Yersinia pestis*, a Gram-negative bacterium, is the causative agent of plague. *Y. pestis* is a zoonotic pathogen that occasionally infects humans and became endemic in the western United States after spreading from California in 1899.
**Methods:** To better understand evolutionary patterns in *Y. pestis* from the southwestern United States, we sequenced and analyzed 22 novel genomes from New Mexico. Analytical methods included, assembly, multiple sequences alignment, phylogenetic tree reconstruction, genotype-phenotype correlation, and selection pressure.
**Results:** We identified four genes, including *Yscp* and locus tag YPO3944, which contained codons undergoing negative selection. We also observed 42 nucleotide sites displaying a statistically significant skew in the observed residue distribution based on the year of isolation. Overall, the three genes with the most statistically significant variations that associated with metadata for these isolates were *sapA*, *fliC*, and *argD*. Phylogenetic analyses point to a single introduction of *Y. pestis* into the United States with two subsequent, independent movements into New Mexico. Taken together, these analyses shed light on the evolutionary history of this pathogen in the southwestern US over a focused time range and confirm a single origin and introduction into North America.

## INTRODUCTION

Given contemporary global trade and travel, understanding the movement and evolution of etiological agents of disease is increasingly important. One such disease is the plague, which is caused by *Yersinia pestis*. It is accepted that *Y. pestis* is a clone of *Yersinia pseudotuberculosis*, with the earliest known *Y. pestis* isolate predicted to have existed 5,700–6,000 years ago (*Rascovan et al., 2019*). *Y. pestis* is a zoonotic bacterium that most often spreads from infected rodents to humans by flea bites. This rod-shaped pathogen is Gram-negative, non-motile, and a member of the *Enterobacteriaceae* family (*Perry &*

Corresponding author
Richard A. Robison,
richard_robison@byu.edu

*Fetherston, 1997*). The *Y. pestis* chromosome is approximately 4.6 megabases in length and is accompanied by three virulence plasmids: pPCP1 (~9,610 bp), pCD1 (~70,500 bp), and pMT1 (~100,980 bp) (*Hu et al., 1998*; *Parkhill et al., 2001*). Plague in humans can manifest itself in three accepted and anatomically distinct clinical forms: bubonic, pneumonic, and septicemic. If diagnosed quickly, common antibiotics such as streptomycin, tetracycline, and chloramphenicol can all effectively treat the plague (*Galimand, Carniel & Courvalin, 2006*). In contrast, without antibiotic treatment, the pneumonic and septicemic forms of the plague are almost always fatal. For this reason, *Y. pestis* is thought to be one of the most dangerous bacterial pathogens for humans (*Croucher et al., 2013*). It is also classified as a Tier 1 select agent by the CDC due to its high fatality rate, ease of growth, and the resultant psychological impact if used as a bioweapon.

Historically, there have been three known *Y. pestis* pandemics. The plague of Justinian, started in 541 AD, is thought to be the first pandemic (*Zietz & Dunkelberg, 2004*). The second plague pandemic occurred from 1347–1800's in Europe and included the "Black Death". During the 5-year period 1347–1352, some studies approximate 15–23 million Europeans, or about 30% of Europe's population, were killed by this pathogen (*Raoult et al., 2013*). The third and final pandemic is hypothesized to have originated in China around 1855 AD (*Zhou, 2004*). Though better sanitation and modern antibiotics have decreased the incidence and spread of *Y. pestis* (*Wagner et al., 2010*), this bacterial species continues to impact human health in various regions around the globe.

Between 1990 and 2006, there were 38,310 reported cases of plague in humans, resulting in 2,845 deaths in over 25 countries (*Galimand, Carniel & Courvalin, 2006*). According to the World Health Organization, there were over 3,248 reported human cases of plague and 584 deaths between 2010 and 2015 (*WHO, 2017*). Although *Y. pestis* is present on every continent except Antarctica and Australia, the foci of endemicity cover only a small percentage (6–7%) of land area (*Anisimov, Lindler & Pier, 2004*). The three highest endemic countries are the Democratic Republic of Congo, Peru, and Madagascar. The latter country had its first introduction of *Y. pestis* in 1898, with 13,234 reported cases from 1998 to 2016 (*Andrianaivoarimanana et al., 2019*; *Morelli et al., 2010*).

The first documented case of plague in North America was in San Francisco in 1899 (*Morelli et al., 2010*), with subsequent spread primarily across the western states. Medical records between 1900–2012 report 1,006 cases of human plague in North America (*Morelli et al., 2010*), with a particularly high concentration of cases in New Mexico, Colorado, and Arizona from 1965 to 2012. The main reservoir of *Y. pestis* in North America is fleas parasitizing prairie dogs, although plague can also be transferred to domestic animals directly from infected prairie dogs (*Rust, 1971*). Exposure to infected felines has been known to lead to infection in humans through bites, scratches, or other contact (*Gage, 2000*).

The relatively recent introduction of *Y. pestis* to the United States is evident in the low genetic and phenotypic diversity of the strains collected in the country. This lack of diversity arises presumably from isolates having undergone a founder effect, which limited diversity introduced in the US (*WHO, 2017*). There is also low sequence diversity in the *Y. pestis* virulence plasmid in North American isolates (*WHO, 2017*).

Evidence for spatial diversity is still observable despite the relatively recent introduction of *Y. pestis* to North America. A single nucleotide polymorphism (SNP) analysis of sequencing data from 34 isolates collected from the western United States revealed an additional 40 previously uncharacterized variants (*Gibbons et al., 2012*; *Lowell et al., 2015*). Subsequent analyses suggest localized genotypic persistence plays a larger role in diversity. Although many *Y. pestis* genetic studies rely on only a few isolates from North America, approximately one-half of the total *Y. pestis* cases in North America occur in New Mexico (*Center for Disease Control and Prevention, 2022*).

Prior studies have focused on genomic variation in subpopulations of *Y. pestis* across various geographical regions. One subpopulation comparison showed that mutations in *Y. pestis* were associated with specific geographical regions and climates in Madagascar (*Andrianaivoarimanana et al., 2019*). However, studies on the evolution of *Y. pestis* within North America have been limited by sample size and relatively narrow time periods that could limit observable patterns of mutation in response to time and selection pressure.

The goal of this study was to analyze and compare the evolution of 22 novel *Y. pestis* genomes collected in New Mexico over a 32-year period, and to interpret these patterns in the context of their geographic and host isolation metadata. Novel *Yersinia pestis* isolates from New Mexico were chosen because of the range of isolate collection years. By performing computational analyses of selection pressure, and reconstructing phylogenetic relationships, we aimed to provide a clearer understanding of the evolution and maintenance of *Y. pestis* in New Mexico in the context of other existing genomes from North America and other geographical regions. The results from such a study could improve our understanding of the evolutionary patterns of *Y. pestis* across a limited region and time frame.

# MATERIALS AND METHODS

## DNA preparation and sequencing

Portions of this text were previously published as part of a thesis (*Warren, 2022*). *Y. pestis* isolates were obtained from the New Mexico State Department of Health prior to characterization and storage at −80 °C in the Biosafety Level 3 (BSL-3) laboratory at Brigham Young University. The 22 isolates used for this analysis were collected between 1983 and 2015 (Table S1). All of the isolates met the selection criteria that included isolation during the desired period of time and possession of all three virulence plasmids (*i.e.*, pCD1, pMT1, and pPCP1) (*Stewart et al., 2008*). Metadata for the selected isolates included year of isolation for all isolates, host species of isolation for 17 isolates, and location of isolation at the county level for six isolates (Fig. S1). All isolates were grown on blood agar in the BSL-3 lab, no contamination was observed, and single colonies were selected for growth. Genomic DNA was then extracted and purified using a MagnaPure LC system (Roche Molecular Systems, Inc., Pleasanton, CA, USA).

Genomes were fragmented using the Covaris M220 focused-ultrasonicator (Covaris, Woburn, MA, USA). DNA cleaning was done with the DNA-Clean and Concentrator Kit (Zymo Research, Irvine, CA, USA). Ends were repaired with NEBNext Ultra II End Repair/dA-Tailing module (NEB, Ipswich, MA, USA) and adapters were ligated using
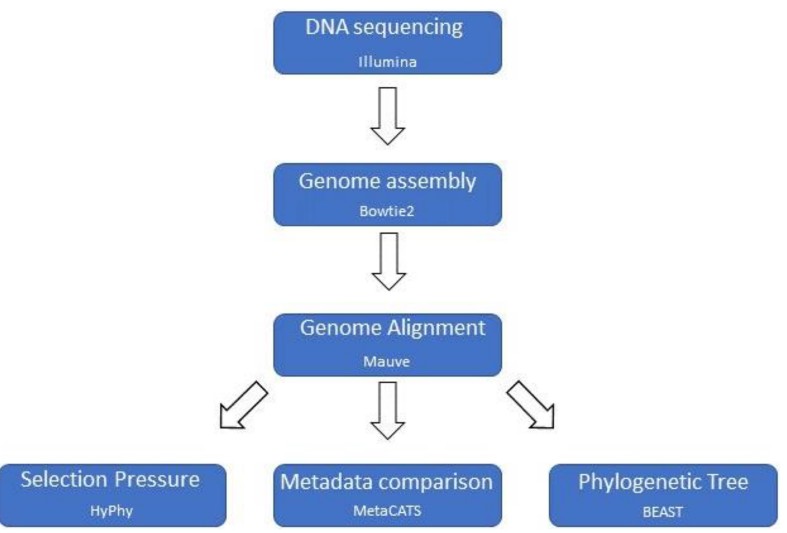

**Figure 1 Computational pipeline outline.** Basic computational pipeline with the core programs. The file formatting scripts are not shown to improve clarity.

NEBNext Ultra II Ligation Module (NEB, Ipswich, MA, USA). Size selection was done with solid-phase reversible immobilization (SPRI) beads followed by KAPA Library Amplification Kit (ROCHE, Indianapolis, IN, USA). The target DNA fragment size of ~500 bp was then verified by gel electrophoresis. Quantification of the DNA fragments was performed using Qubit fluorometry (Thermo Fisher, Waltham, MA, USA). The final DNA fragments were sequenced at the University of Pittsburgh, Genomics Research Core (Pittsburg, PA, USA). Isolates were sequenced with Illumina MiSeq v3 chemistry, using the paired-end 150 cycle kit to generate 22–25 million total reads.

A custom computational pipeline was constructed to assemble genomes, perform a multiple genome alignment, and parse the data for selection pressure analysis, metadata association analysis, and phylogenetic tree reconstruction (Fig. 1).

## Genome assembly and annotation

Bowtie2 version 2.4.1 default settings was used to generate a reference-based mapping of each novel genome from the fastq files (*Langdon, 2015*), using *Y. pestis* strain CO92 as the reference genome (GenBank accession: AL590842.1). The read assembly for the chromosome and each of the three plasmids for each genome was then converted to a consensus fasta file using samtools version 1.11 and bcftools version 1.11. A limitation to mapping reads to the reference is that reads are placed where they best match the reference genome. Consequently, we anticipated difficulty inaccuratelying identify which version of a multi-copy gene contained any observed variation. The depth of coverage for the 21 newly-sequenced isolates was greater than 30, with isolate 1880 having the minimum depth of 9.48×. Average breadth of chromosome coverage was 99% with an average depth of 41.5 (Table S2). The average number of SNPs for each novel *Y. pestis* genome was 42.7. A *de novo* assembly was performed on two genomes and showed no variation from the reference-based assembly, as such the reference-based method was used for all genome
assemblies. The plasmids pPCP1, pPCD1, and pMT1 had coverage depths of 500×, 200×, and 165×, respectively. Plasmid coverage breadth was 0.99, 0.99, 1, for pPCP1, pPCD1, and pMT1 (Tables S3 and S4). Mauve was then used to create a synteny-independent multiple genome alignment of the newly assembled *Y. pestis* isolates (*Darling et al., 2011*). Annotation was performed using the Rapid Annotation using Subsystem Technology (RAST) server (*Aziz, Bartels & Best, 2008*).

To further our confidence in the alignments, Snippy (version 4.6.0) was used for variant calling on all genomes and plasmids with default settings (*Seemann, 2020*) with the above mentioned reference genome (CO92; GenBank accession: AL590842.1). Once consensus genome sequences were generated with Bowtie2, the variants in the consensus genomes were compared to the variants from the Snippy analysis. MEGA (version 11) (*Tamura, Stecher & Kumar, 2021*) was used to calculate the percent identity of the two genomes. All chromosome genomes had a pairwise identity of 99.99% and all plasmids had a 100% pairwise identity. The Bowtie2 consensus sequences were consequently used for all further analyses.

## Metadata association analysis

The metadata-driven comparative analysis tool for sequences (meta-CATS), was with to identify sequence residues displaying significant skew among two or more groups of sequences using a chi-square statistic on nucleotides (*Pickett et al., 2013*). Three statistical analyses were performed based on the host or time of isolation for the *Y. pestis* sequences including (1) human hosts *vs* non-human host, (2) feline host and non-feline host (human and unknown), and (3) time of isolation across five-year time periods. Limited sequence variation based on the host of isolation was expected as felines and humans are not the main reservoirs, but we assumed that most evolutionary patterns should arise when comparing year of isolation metadata. To simplify the analysis, two scripts were used to automate the extraction of the non-conserved positions of the chromosome and plasmids among all *Y. pestis* isolates prior to meta-CATS analysis. Specifically, the order-Homol-Chunk.py script takes the MAUVE alignment as input and returns a text file of homologous positions of the chromosomes, regardless of their synteny. The parseNonConservedSeq.py script returns a text file with the aligned homologous positions and sequence variations, which was defined as two or more sequence variations present in at least three genomes. A multiple sequence alignment was constructed to enable downstream comparisons of homologous positions. First, this alignment was used to identify nucleotide sequence variations that were significantly associated with metadata attributes including year or host of isolation. Specifically, the meta-CATS algorithm was used to compare *Y. pestis* sequences from known human hosts to non-human hosts (*e.g.*, unknown hosts or feline hosts). A second meta-CATS analysis was performed to identify differences in isolates collected from feline hosts with those from non-feline hosts (*e.g.*, unknown hosts or human hosts). A third comparison was performed to examine the three virulence plasmids of *Y. pestis* in groups by year of isolation. The fourth and final meta-CATS analysis utilized the full genomes grouped by their year of isolation (Table S5).

## Selection pressure analysis

The software package HyPhy (*Pond, Frost & Muse, 2005*) was used to detect evidence of selection using the Fixed Effects Likelihood (FEL) and the Mixed Effects Model of Evolution (MEME) methods (*Pond & Frost, 2005*). FEL uses a maximum likelihood approach to infer nonsynonymous and synonymous substitution rates on a per-site basis, thus predicting which genes are under positive or negative selection. FEL assumes all sites undergo constant selection pressure, which meets the assumptions from the observed regional variations noted from the sequences included in this study. MEME performs a mixed-effects maximum likelihood approach to test whether individual sites have been subject to episodic or diversifying selection (*Murrell et al., 2012*). These analyses were performed separately on the chromosome and the pCD1, pMT1, and pPCP1 virulence plasmid sequences of the *Y. pestis* isolates. We intentionally selected this approach to enable the detection of different evolutionary forces acting on plasmids and the chromosome. To calculate the percent similarity and simultaneously exclude the possibility that multiple loci in the genomes contain the same sequence, the four genes predicted to have undergone negative selection were also compared using BLASTN (*Sayers et al., 2022*).

## Phylogenetic reconstruction

Twenty-two novel sequences from New Mexico (this study), together with 16 genomes from the Pathosystems Resource Integration Center (PATRIC) database, and the CO92 genome from the National Center for Biotechnology Information (NCBI) were included in this study. The alignment was created using Mauve (*Darling et al., 2011*). BEAUti software was used to prepare tip-dated files prior to phylogenetic reconstruction using BEAST2. BEAST2 was run with default settings, relaxed exponential clock, substitution model JC69, priors were set to yule model, and dates of isolation were used as tip dates (*Bouckaert et al., 2019*). The phylogenetic tree was reconstructed from 50 or 10 million iterations, with logging every 1,000 samples. The effective sample size (ESS) is the number of independent draws from the posterior distribution. All of the ESS calculated values met acceptable criteria, defined as greater than 100, using the Tracer program (*Rambaut, 2021*). The first 1,000 trees (10% of samples), were discarded as burn-in prior to visualization with FigTree (*Rambaut, 2021*).

To root the New Mexico tree and reconstruct phylogenetic relationships with greater confidence in the homology statements (alignment), two additional trees were built using the same parameters and pipeline as above. The first tree used the 22 New Mexico genomes along with 16 additional genomes collected from Russia, China, USA, and New Mexico (downloaded from PATRIC) and rooted with *Yersinia pseudotuberculosis*. This tree reconstruction allowed a focused view of the New Mexico isolates together with other isolates from the United States. The second reconstruction utilized a subset of taxa from the first analysis. This second analysis included known isolation hosts and utilized the original 23 genomes and 22 novel New Mexico genomes, along with the 632.720| CP064127.1 genome sequence downloaded from PATRIC.
### Co-evolutionary analysis

A more in-depth analysis of phylogenetic co-evolution was performed for each virulence plasmid and bacterial chromosome using Jane (*Conow et al., 2010*). Jane performs tree reconciliation in the context of biological events and associated costs. The input required is a nexus file that includes a host tree, a parasite tree and a list of pairs of host: parasite relation. We used the default weighted scores for five types of events: co-speciation, duplication, duplication with host switch, loss, and failure to diverge.

## RESULTS

### Metadata associated analysis

We first wanted to determine whether any variants were significantly associated with the host of isolation. As such, two sets of comparisons were performed between chromosomes obtained from bacteria infecting (1) human hosts *vs* those infecting non-human hosts and (2) feline hosts *vs* non-feline hosts. We observed no positions that surpassed the significance threshold in these first two meta-CATS analyses. We then performed a third analysis that examined the three virulence plasmids in groups by year of isolation. Each of these temporal groups were created from sequences isolated in a similar range of years and used natural gaps between the range of years to make the distinction between groups. In this analysis, we observed no significant positions in plasmids pMT1 or pPCP1, and one significant position on the pCD1 plasmid. The significant position on pCD1 is in the *yscP* type III secretion gene at base 278, which varied between adenine in groups 1–3 and cytosine in group 4 (*p*-value = 0.046) resulting in a nonsynonymous change from histidine to proline.

We next wanted to determine whether there were any variants that were significantly associated with the time of isolation. Consequently, we performed a separate meta-CATS analysis after grouping the genomes into four temporal groups by the year of isolation (four isolates from 1983–1988, two isolates from 1993–1998, five isolates from 2003–2009, and 11 isolates from 2011–2015). The analysis identified 42 significant nucleotide sites across 15 chromosomal loci with *p*-values < 0.016 (Table S5). Six of these 42 sites were located in genes annotated as hypothetical proteins. Nucleotide positions 552580 and 4384651 were located in a tRNA and the 5S-rRNA coding region, respectively. We also observed seven statistically significant sites that varied by temporal group in the *sapA* gene, a peptide ABC transporter. The coding region for a ferric iron ABC transporter, PSF113_5656 had three sites of significant temporal variation. The *fliC* gene had nine sites that were temporally significant. The coding region for the *argD* enzyme, a catalyst in the lysine biosynthesis pathway, had six notable temporally-associated SNPs; while the IS100 inverted repeat sequence contained multiple significant sites. Overall, this analysis revealed 45 sites, 42 of which were in the chromosome and three sites in the virulence plasmids, that showed noticeable patterns when the temporal metadata was included. Interestingly, we found three genes that together contained 22 sites that significantly associated with the time of isolation (Table 1).

**Table 1 Meta-CATS analysis.**

| Groups | ArgD | FliC | SapA |
|---|---|---|---|
| 1983–1988 [4] | ———— [4] | AGGGGGGTA [4] | ACATCAC [4] |
| 1993–1998 [2]* | ———— [2] | ————————— [2] | ———— [2] |
| 2003–2009 [5]* | GAAAGC [4] <br> ———— [1] | AGGGGGGTA | ACATCAC [4] <br> ———— [1] |
| 2011–2015 [11]* | GAAAGC [10] <br> ——C-[1] | AGGGGGGTA [10] <br> ————————— [1] | ACATCAC [11] |
| P-value | 0.0065 | 0.0026 | 0.0017 |
| CO92 chromosome position nucleotide (codon number) | 190085–190090 (63361–63363) | 796168–796176 (265389–265392) | 2647532–2647538 (882510–882512) |

Note:
Meta-CATS top results across three genes, *ArgD, FliC,* and *SapA*.
* Numbers in brackets represent the number of genomes in each group. X represents a gap '—' in the sequence alignment.

**Table 2 Significant HyPhy results.**

| Genome | Position | Identical sequences | Genome | Number of sites | Function |
|---|---|---|---|---|---|
| Chromosome | 2304034 | All plasmids | + | 1 | Transposase IS1541 |
| Chromosome | 2451043 | pMT1 | − | 1 | Transposase IS100 |
| Chromosome | 4434982 | None | − | 4 | Invasin |
| pCD1 | 29966 | None | + | 1 | Type III secretion |

Note:
Negative pressure results from HyPhy with position, if there are identical sequences, direction, number of sites, and functions.

## Selection pressure (HyPhy)

We next sought to determine whether specific sites across the genome were subjected to selection pressure, calculated from proportions of nonsynonymous substitutions and synonymous substitutions in aligned codons. To do so, we specifically focused our analysis on the Fixed Effects Likelihood (FEL) and the Mixed Effects Model of Evolution (MEME) algorithms within the HyPhy software. We used MEME to detect either episodic or diversifying selection but observed no significant results. Similarly, we detected no regions under positive selection using the FEL algorithm. However, we did identify four different genes that contained at least one codon predicted to undergo negative selection (Table 2).

Chromosome locations 2304034 and 2451043, which are part of a transposase or mobile element protein, were detected as significant in this analysis; as was location 4434982, which is located in an invasin gene that contained four sites with evidence of negative selection within codons 1099, 1106, 1126, and 1321. In pCD1, there was one site of negative selection within a type III secretion gene (*yscP*). The pMT1 and pPCP1 plasmids had no coding regions that demonstrated selection pressure.

We then used BLAST to confirm the percent similarity and exclude the possibility of multiple matching loci in the genomes for these regions that were predicted to have undergone selection pressure. The results of this analysis showed all three coding regions in the chromosome (transposase, mobile element, and invasin protein) were identical to those of other *Y. pestis* isolates as well as to a subset of *Y. pseudotuberculosis* isolates.

The mobile element protein sequence was found in all of the virulence plasmids (*i.e.*, pCD1, pPCP1, and pMT1).

The transposase sequence at nucleotide position 2304034 is identical at the amino acid level to sequences in all three virulence plasmids. The mobile element protein at position 2451043 had an identical portion in pMT1. There were also 64 sequences in the chromosome with a variation of only one nucleotide. The invasin and type III secretion regions on the chromosome and pCD1 plasmid, respectively, had no similar sequences.

## Phylogenetic analysis

Lastly, we wanted to perform multiple Bayesian phylogenetic reconstructions, which estimates rooted time-measured phylogenies with a molecular clock, between our New Mexico sequences and those from other geographical regions. To do so, we concatenated the aligned locally co-linear blocks (*i.e.*, homologous regions) contained within the MAUVE alignment of various *Y. pestis* genomes with regions having no homology to other genomes being represented with gaps. Our first reconstruction combined novel New Mexico genomes, CO92, *Yersinia pseudotuberculosis*, and 41 *Y. pestis* sequences from the PATRIC database to determine the optimal outgroup with 500 million iterations (See Table S6 for isolate collection year and origin). Given that *Y. pestis* evolved from *Y. pseudotuberculosis* (*Demeure et al., 2019*), we used the latter as the initial global outgroup taxon to both root the initial and to aid in rooting subsequent trees (Fig. 2A). Using this preliminary tree, we then reconstructed a subsequent tree using the novel New Mexico isolates, CO92, and all PATRIC sequences with recorded year of isolation (Fig. 2B). This tree was rooted with the Russian 632.720_R sequence, shown in Fig. 2A as sister to the rest of the *Y. pestis* isolates. The resulting tree shows a monophyletic pattern among the USA isolates but no apparent pattern of host or isolation year. We then reconstructed a separate tree, which focused specifically on isolates collected from the United States. This tree included all novel New Mexico sequences with the 632.715 sequence from the United States as the root. This reconstruction revealed variation in host-pathogen relationships. All but three nodes in this tree had posterior probability values greater than 0.97 (Fig. 3).

Trees were also reconstructed using sequences from each of the three virulence plasmids; however, due to their smaller size and sequence conservation, all three of these trees lacked sufficient node support to confidently infer topological resolution. Effective sample sizes for the plasmid trees were high, with values of 3,972.5, 3,817.5, and 2,937.5, for pMT1, pCD1, and pPCP1, respectively. The range of posterior probability scores, a BEAST metric that represents the conditional probability within a range of 0.00–1.00 calculated, was low: pMT1 (0.15, 0.48), pCD1 (0.10, 0.44), and pPCP1 (0.01, 0.11). Unsurprisingly, JANE co-evolutionary analysis of plasmid sequences produced no significant results.

## DISCUSSION

The purpose of this study was to identify patterns of evolution in *Y. pestis* genomes collected over a 32-year period in New Mexico. Though our study focused on a relatively small-temporal window, prior studies have revealed genetic variation in a single plague

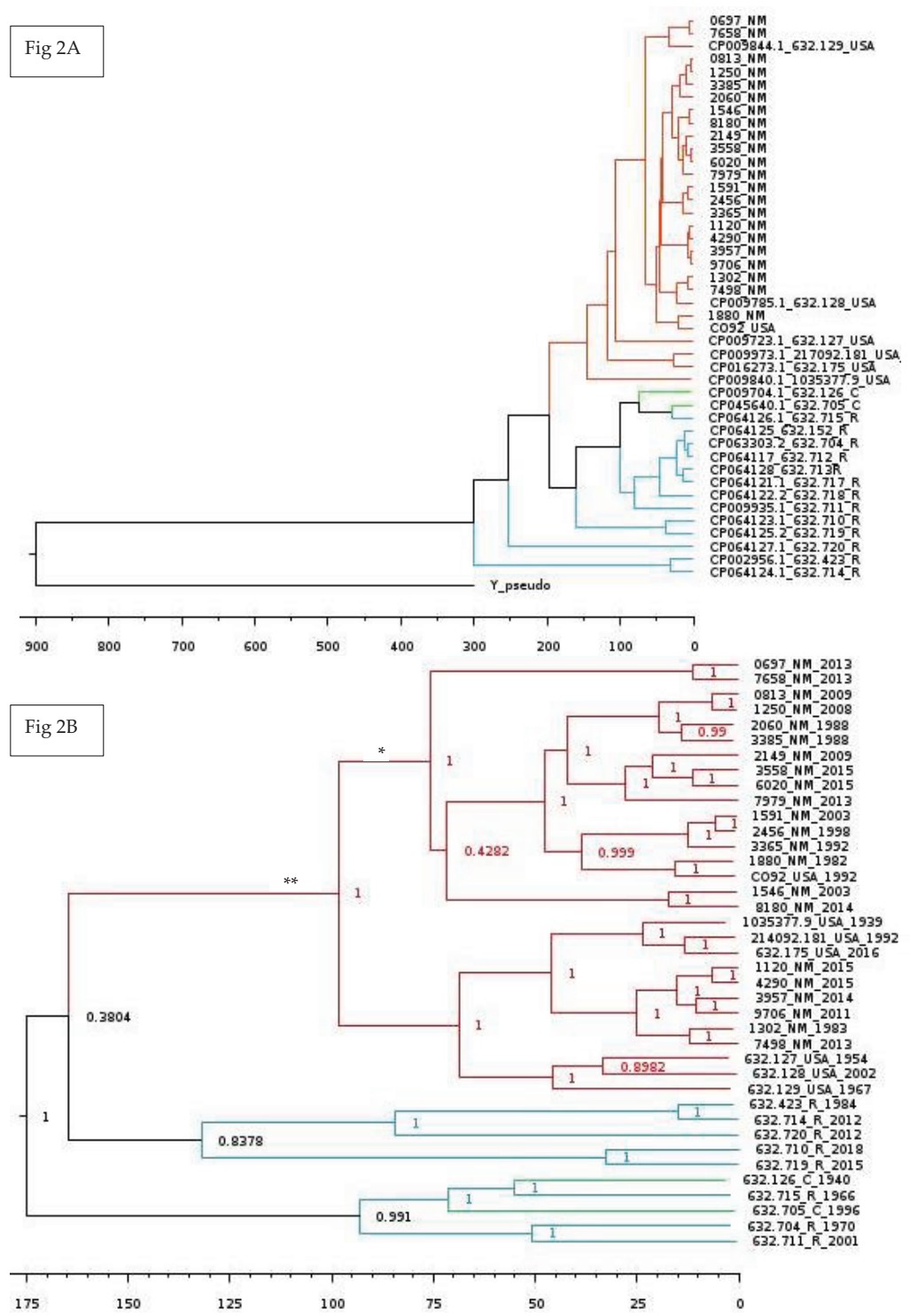

Fig 2A

Fig 2B

**Figure 2 Global Bayesian tree reconstruction of *Yersinia pestis* isolates.** (A) Bayesian tree reconstruction of *Yersinia pestis* isolates with *Yersinia pseudotuberculosis* as an outgroup using the substitution model JC93 and 50 million iterations. All posterior probability values are higher than 0.83 except one (0.43). The effective sample size (ESS) is 365. Twenty-two novel *Y. pestis* genomes from New Mexico are included along with 20 genomes from the PATRIC database. Genomes isolated from Russia, China, USA, and New Mexico are indicated by the following suffixes at the end of the strain identifier: R, C, USA, and NM, respectively. Colors represent region of isolation; red, green, and blue represent isolates from the

**Figure 2 (continued)**
United States, China, and Russia, respectively. Branch lengths are scaled by time. (B) Bayesian tree reconstruction of New Mexico and USA Isolates with the substitution model JC93. A total of 22 New Mexico isolates and an additional 16 publicly available isolates from the PATRIC database were included. The ESS value is 130. Posterior probabilities are provided at their respective nodes. Genomes isolated from Russia, China, USA, and New Mexico are indicated by the following suffixes at the end of the strain number: R, C, USA, and NM, respectively. Colors correspond to regions. Year of isolation is included at end of genome name. The ** indicates the branch that contains only and all isolates from America. The * indicate putative introductions into New Mexico. Branch lengths are scaled by time with the scale bar at the bottom representing the divergence times in years since present. This tree suggests the evolution of the New Mexico isolates from *Yersinia pestis* isolated from Russia and China. More information is needed to determine exact dates of introduction to various regions.

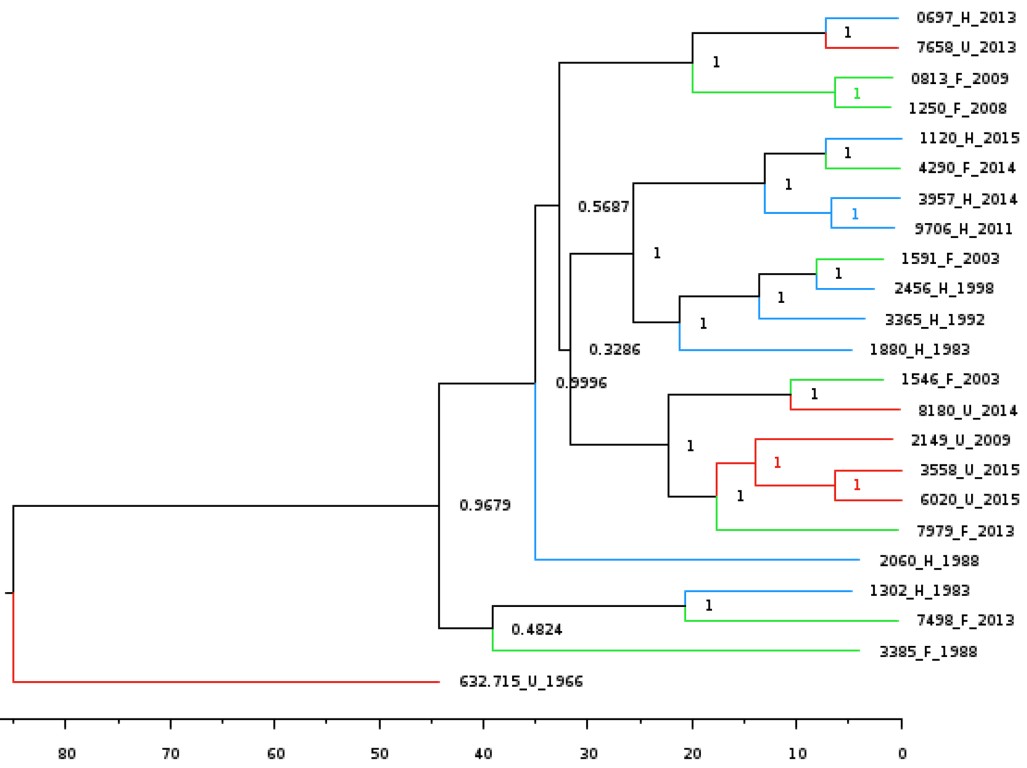

**Figure 3 Bayesian phylogenetic tree reconstructed using the 22 novel New Mexico genomes rooted with 632.715.** Posterior probabilities are provided at their respective nodes. The effective sample size (ESS) was 618. Relevant metadata are appended to the sequence names to reflect human (H) or feline (F) host of isolation along with year of isolation corresponding to blue and green lines respectively. The JC93 substitution model was used and branch lengths are scaled by time. The scale bar at the bottom represents the divergence times in years since present.

season and small regional area specifically identifying 39 new putative SNPs, indels or tandem repeats among isolates with nearby geographical locations of isolation (*Gibbons et al., 2012*). Though a reference-based assembly could result in missed genes, comparing a multiple sequence alignment of the reference-based assembly to two *de novo* assemblies showed no variation. We observed no evidence of positive or episodic selection pressure in our analyses, confirming previous findings (*Conow et al., 2010*). However, regions in four genes indicated the presence of negative selection. We also identified 43 sites of nucleotide
variants that significantly correlated with the isolation period. In addition this study reports the phylogenetic relationships of multiple genomes that depict monophyly of USA isolates but with no consistent pattern between topology, geographic location, and host.

## Metadata-associated sequence analysis results

Our analysis associating nucleotide sequence positions to metadata yielded 42 statistically significant sites. This analysis is most sensitive at identifying genomic loci that significantly differ between groups of sequences when more sequence diversity is present. Although the precise reason that the extant sequences are more divergent from their ancestor sequences is unclear, at least one possible explanation could be due to sampling bias. We focused on the three regions of the genome with the most significant associations with isolation year.

The first genomic locus that significantly differed was the *fliC* flagellin gene, which contained nine consecutive sites with significant sequence variants between temporal periods. Flagellin plays a role as an immunogen in host humoral immune responses (*Honko et al., 2006*). More specifically, FliC is a flagellar filament structural protein and a Toll-like receptor five agonist. Prior studies have focused on FliC in potential vaccine development (*Honko et al., 2006*). Our results, when compared with isolation year, indicate a three-codon deletion in isolates collected between 1993 and 1998 that was only present in one of the later 16 isolates. Several mutations, including mutations in the FlhD regulator protein inhibit expression of flagellar genes (*Bates et al., 2008*). Future experiments are needed to better understand the processes driving this selection pressure and the potential implications.

The second genomic locus that displayed a significant association between sequence positions and the temporal period of isolation was *sapA*. This gene is located on the *Y. pestis* chromosome at positions 2647475–264912118 and codes for a peptide ABC transporter substrate-binding protein. The SapA protein has a relative affinity to bind heme, which is like the heme-binding DppA protein in *E. coli* (*Bates et al., 2008*). Similar functions are seen in HemTUV, which is involved in high-affinity heme transport. When inactivated it does not inhibit the ability of *Y. pestis* to uptake heme. The acquisition of certain ions is key to the successful pathogenesis of *Y. pestis* (*Mason et al., 2011*). Our study identified a seven-base deletion in the middle of the *sapA* gene, and a potential frameshift. Isolates collected during earlier and later years had sequence information, while the middle years of isolation lacked the seven bases. This pattern could indicate that the deletion was not favorable in the New Mexico isolates and thus was not as prevalent in later populations.

The final genomic locus that contained significant sequence variation was in the *argD* gene, located at chromosome positions 189518–190123, which was not available in the original sequence annotation, but was predicted with the RAST server. The ArgD enzyme catalyzes the sixth of nine steps in the lysine biosynthesis pathway and is also involved in arginine biosynthesis. However, it has been shown that the lack of functional ArgD does not cause auxotrophy (*Bearden, Staggs & Perry, 1998*). Though this plays a specific role in lysine biosynthesis, the lack of auxotrophy in deletion mutants suggests that there is at least one alternate metabolic pathway for lysine biosynthesis in *Y. pestis*. The effect of the 6 bp
deletion on ArgD protein activity is unknown and should be further characterized to better understand whether it has any phenotypic effect. In our metadata analysis, the two-codon region showed a variant involving the presence and absence of genetic material. The two earliest groups of sequences by year of isolation, 1983–1998, had sequence gaps while the same region in later genomes had nucleotides sequences, with only two of 16 resulting in gaps. This difference could indicate a transition from the lack of this sequence towards a two-codon insertion.

## Predicted selection pressures

This study revealed codons in four genes that were predicted to be under negative selection. The first is the IS1541 transposase sequence, located at chromosomal position 2304034–2304492. A previous study on *Y. pestis* showed this taxon had several IS element groups, including IS100, IS285, and IS1541 (*Charusanti et al., 2011*). IS1541 is an insertion element that contains a single open reading frame that is known to play a part in the ability of *Y. pestis* to escape the host immune system (*Odaert et al., 1998*). Structurally IS1541 is similar to the IS200 which is found in *Salmonella enterica* (*Charusanti et al., 2011*). The genome annotation for *Y. pestis* shows multiple identical genes and gene products in all three virulence plasmids. At least four different insertion sequences have been found previously in the chromosome, along with 66 complete or partial copies of IS1541 (*Cornelius et al., 2009*).

The *y1093* gene, which is located at chromosomal position 2451043–2452065 and is a transposase for insertion sequence IS100, also contained codons showing evidence of negative selection pressure. The reference genome, *Y. pestis* CO92, potentially contains more than one hundred copies of IS100, IS285, and IS1541 (*Devalckenaere et al., 1999*). This agrees with our findings of 43 near-identical copies in the chromosomes and two copies on the pMT1 plasmid, with the pMT1 copies present on opposite strands (*Motin et al., 2002*).

The third gene containing codons predicted to undergo negative selection was locus tag *YPO3944*, which is located on the *Y. pestis* chromosome at position 4434982–4444023. This gene codes for an intimin/invasin-like protein and has structural similarity to the invasin and intimin proteins found in *Y. pseudotuberculosis*. The YPO3944 gene product has two predicted components; the first is a repetitive bacterial immunoglobulin domain, while the second is a lysin motif (LysM) (*Hu et al., 1998*). Prior work showed this protein localized to the outer membrane and that deleting the gene did not impact bacterial infection in fleas (*Seo et al., 2012*).

The final gene that contained codons predicted to be subjected to negative selection was the *yscP* gene in the pCD1 plasmid, which is part of the YscP type-three secretion system (T3SS) of *Y. pestis* (*Seo et al., 2012*). The low-level calcium-regulated YscP protein, which is part of the Ysc injectisome (*Edqvist et al., 2003*), is one of the determining factors in the length of the injectisome needle, and works in conjunction with the YscO protein (*Stainier et al., 2000*). Additional experiments are needed to better understand the role of the *YPO3944* and *yscP genes* when under selection pressure.

Our identification of four codons within the *YPO3944* gene that are under negative selection warrants further study. The YscP protein is a part of the T3SS, which has a critical function in *Y. pestis* pathogenesis. The single gene copy of the invasin and the *YPO3944* gene is consistent with the idea of low genetic diversity among *Y. pestis* isolates in North America. A prior study showed positive selection of the *rpoZ* gene during colder and drier periods that our study did not detect (*Cui et al., 2020*). This may be due to the limited regional variation, limited environmental changes, different phylogenetic lineages, and decreased sequence diversity of *Y. pestis* genomes included in the analysis. Though evidence of selection pressure is shown in our study, further studies are needed to determine physiological and biological importance of all significant variations.

## Interpreting phylogenetic reconstructions

A total of 43 genomes were used in the various phylogenetic reconstructions, including the 22 novel *Y. pestis* isolates collected from New Mexico. Tree reconstruction involved the novel sequences from New Mexico, CO92, 20 genomes downloaded from PATRIC, and *Y. pseudotuberculosis* as the outgroup. From this initial tree the monophyletic pattern of these isolates supports the hypothesis of a shared common ancestor for plague isolates in New Mexico and areas in the USA, supporting prior studies (*Gibbons et al., 2012*; *Morelli et al., 2010*). More specific regional information from all USA isolates obtained from PATRIC is needed to further elucidate interleaved patterns. Our findings are also consistent with the assertion that *Y. pestis* spread eastward after initially being introduced to California in the late 1800's (*Demeure et al., 2019*). It is important to recognize the difference between introduction into a given geographical area and evolutionary emergence. The former still maintains much of the ancestral sequence, while the latter infers a unique set of genetic characteristics that distinctly separates the new sequence from its ancestors. The second tree had all the same genomes but lacked the *Y. pseudotuberculosis* sequence (Fig. 2B).

Tree reconstruction using our set of 22 novel New Mexico genomes over 32 years, which used the 632.715 genome as the ancestral outgroup, allowed a more precise comparison of host variation and isolated regional time results. The spontaneous substitution rate for *Y. pestis* has been estimated at $2.9 \times 10^{-9}$ to $2.3 \times 10^{-8}$ nucleotide changes per site per year (*Demeure et al., 2019*). Though substitution rates vary across lineages (*Cui et al., 2013*), limited regional variation on these sequences reduces the number of genetic lineages. The tree reconstructed using our novel genomes had extremely high posterior probability values, providing a highly resolved glimpse into the evolutionary history of *Y. pestis* in this geographical and temporal period. The topology of this tree reconstruction indicates two independent dispersal events into New Mexico. At least one potential explanation is that populations are genetically stable within geographic areas, and the prevalence of genotypes in given years surges when those localized populations expand during regionally constrained epizootic events (*Vogler, 2017*). Distribution of many of the genotypes is consistent with the phylogenetic analysis. Coevoluationary analyses of chromosome and virulence plasmids confirm a lack of significant association between genome host or year of isolation.

The final phylogenetic reconstruction involved the relatively few dissimilar virulence plasmids from our novel New Mexico genomes (*Hu et al., 1998*). Past studies have shown evidence of horizontal gene transfer involving genes on each of these plasmids (*Sun, Hinnebusch & Darby, 2008*). However, our limited sample size as well as limited geographic and temporal diversity resulted in insufficient variation to confirm such a phenomenon within these plasmids.

## CONCLUSIONS

In summary, this study depicts the evolution of *Y. pestis* over a 32-year temporal period within a limited geographical region. Our phylogenetic analyses support a single introduction of *Y. pestis* into the United States, and subsequent evidence of negative selection in four genes in the *Y. pestis* New Mexico genomes. The two chromosomal genes, IS100, IS1541, and the pCD1 gene *Yscp*, each had a single site of negative selection, while the fourth gene, YPO3944, had four codons undergoing negative selection. Our 22 newly sequenced isolates from New Mexico showed no statistically significant evidence of positive or episodic selection. However, we did identify 42 positions that displayed statistically significant variation associated with the year of isolation. Phylogenetic tree reconstructions and coevolutionary analyses showed no consistent patterns of host variation or year of isolation.

## ACKNOWLEDGEMENTS

The authors thank the Health Department of New Mexico for supplying the novel *Yersinia pestis* isolates.

### Funding

The authors received no funding for this work.

### Competing Interests

Brett E. Pickett is an Academic Editor for PeerJ.

### Author Contributions

- Mary E. Warren conceived and designed the experiments, performed the experiments, analyzed the data, prepared figures and/or tables, authored or reviewed drafts of the article, and approved the final draft.
- Brett E. Pickett conceived and designed the experiments, authored or reviewed drafts of the article, and approved the final draft.
- Byron J. Adams conceived and designed the experiments, authored or reviewed drafts of the article, and approved the final draft.
- Crystal Villalva performed the experiments, authored or reviewed drafts of the article, and approved the final draft.
- Alyssa Applegate performed the experiments, authored or reviewed drafts of the article, and approved the final draft.

- Richard A. Robison conceived and designed the experiments, authored or reviewed drafts of the article, and approved the final draft.

## DNA Deposition

The following information was supplied regarding the deposition of DNA sequences:

The sequence reads for the novel New Mexico isolates are available at SRA: PRJNA833046.

The genomes in the PATRIC database are available at the Bacterial-Viral Bioinformatics Resource Center (BV-BRC): https://www.bv-brc.org.

## Data Availability

The code is available at GitHub and Zenodo:

https://github.com/mepWarren/pestis-scripts.

Mary Elizabeth Warren. (2023). Comparative sequence analysis elucidates the evolutionary patterns of *Yersinia pestis* in New Mexico over thirty-two years. https://doi.org/10.5281/zenodo.8193316.

## Supplemental Information

Supplemental information for this article can be found online at http://dx.doi.org/10.7717/peerj.16007#supplemental-information.

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
