# Peer review of "Comparative sequence analysis elucidates the evolutionary patterns of Yersinia pestis in New Mexico over thirty-two years"

_PeerJ, doi:10.7717/peerj.16007_

## Round 0.1 · original submission · Major Revisions

Dear author,

We kindly request that you address the concerns raised by the reviewers regarding the availability of data used in your analysis. It is important to provide links or references to public databases or repositories where readers can access the raw data. We want to assure you that all reviewers agree on the relevance and timeliness of your contribution and encourage you to submit an improved version of your manuscript.

·

Basic reporting

I really enjoyed that the paper was straightforward, concise, and simple to comprehend!
- There are grammatical errors (missing white spaces, switching tenses) and inconsistent intext referencing (line 75). Those sections of the paper may require revision!

Experimental design

- It appears that the US isolates were retrieved from the PATRIC database. It would be beneficial to provide more context for these isolates, such as when they were collected. This would improve the reader's comprehension of the data.

Validity of the findings

- It would be useful to know what the paper's purpose is. Perhaps a couple of sentences in the introduction explaining the authors' expectations before conducting the analysis and why they focused on the plague specifically in New Mexico?
- The results are presented well, but the discussion seems somewhat redundant. It would be useful to the reader to expand on the implications of the results. For example, the paper explains negative selection for four genes. Are there any physiological or biological interpretations of these variants or genes that are undergoing more selection? Why are these genes of particular interest to the author(s)?
- In Fig. 2A, many isolates from New Mexico and the United States are interleaved. Would that be evidence that Y pestis was introduced into the United States from New Mexico?

Reviewer 2 ·

Basic reporting

This study provides valuable insights into the evolutionary history of pathogen Y. pestis in the southwestern United States. The authors focused on four specific genes (sapA, fliC, argD, and Yscp), with sapA, fliC, and argD showing the most statistically significant variations. They concluded that Y. pestis was introduced into the United States only once, with two independent movements into the state of New Mexico. The study is divided into three sections, and the authors conducted separate analyses to support their conclusions.

Introduction:
The introduction provides a concise background on the importance of studying Y. pestis and its origins. The authors discuss the various plagues caused by this bacterium, highlighting the significance of their research. They clearly state the scope of their study, setting the stage for the rest of the paper. However, the reference on line 75 needs to be edited to match the overall format.

Experimental design

Materials and Methods:
The authors provide clear and detailed explanations of each step in the sample collection, extraction, library preparation, and analysis. They discuss three types of analyses (meta-CATS, selection pressure, and phylogenetic reconstruction) that were performed to address the research question. To enhance clarity, it might be useful to mention "Illumina" on line 125 before MiSeq since companies are mentioned for the library prep kits.

Validity of the findings

Results:
In this section, the authors present the findings of their study in a detailed manner. For the metadata-associated analysis, no significant positions/genes were observed when comparing human vs. non-human and feline vs. non-feline hosts. However, the third meta-CATS analysis revealed 42 significant nucleotide sites across four genes. To provide a more comprehensive view, it would be beneficial for the authors to also mention the findings for ArgD in the Metadata Associated Analysis section of the results, considering that FliC and sapA are mentioned.
The other two sections on selection pressure and phylogenetic analysis are adequately detailed and clearly present the findings. It might be helpful to briefly mention the previous finding on line 307 to provide context.

Discussion:
The authors have done an excellent job summarizing the findings and discussing their significance in this section. However, it would be informative to briefly mention the previous finding being referred to on line 307. Additionally, Table 2 shows "+ selection" for pCD1, while line 264 mentions this site as having exhibited negative selection. This discrepancy should be addressed and clarified.

Additional comments

In general, the authors have done an excellent job structuring the study and providing solid data to support their findings. The manuscript is written in a professional and concise manner, showcasing the authors' expertise in the field. I believe this study has the potential to make a significant impact on the scientific community in furthering our knowledge and understanding of Y.pestis. Great job!

Annotated reviews are not available for download in order to protect the identity of reviewers who chose to remain anonymous.

Reviewer 3 ·

Basic reporting

The study by Warren and colleagues describes patterns of evolution in 22 Yersinia pestis genomes collected over a 32-year period from New Mexico, USA, a region with persistent plague reservoirs since the disease’s introduction during the third plague pandemic. While the range of analyses presented may be of potential interest to plague specialists, especially with regard to bacterial adaptation within local environments, the presentation of results and references to previous work are at times insufficient and difficult to follow. Below is a number of comments that should be addressed in a revised version of the paper:

Introduction:

The authors present mortality estimates for historical plague pandemics (first and second pandemics) which are highly debated in plague research. Moreover, the citations used for these estimates are incorrect and do not represent the historical literature. Such statements should be revised, taking into account the most up-to-date historical, archaeological and genetics research on historical plague pandemics.


Results:

Line 89: “Prior studies have focused on the historical worldwide spread of Y. pestis [21] [22]. These studies suggest that Y. pestis originated in China and then crossed continents following historical trade routes.” Here it is unclear what authors mean by “originated”. Does that mean the origin of historical pandemics, or the prehistoric origins of the bacterium? In any case, these statements present oversimplifications of ongoing investigations/debates in the fields of plague evolution and history and should be avoided. Moreover, the citations used are either almost 20-years old or are not related to the topics presented and should be replaced with more suitable ones.


Minor comments:

- The authors should check typos within the entire manuscript., for example, in line 376 “fleas c”.

- Similarly, species names should always be indicated in italics

Experimental design

Phylogenetic analysis:

- With regard to the newly presented genomes, a map of their isolation locations would aid clarity in the phylogenetic results.

- The choice of a strict clock for the Bayesian phylogenetic analysis is surprising given the known variable rate of evolution in Y. pestis. The authors should explain their prior choices as well as their testing strategy for these choices.

- The scale axes in each presented tree should be clearly labelled, as it is unclear what they represent. In some of the figure legends it is mentioned “Branch lengths are proportional to the number of nucleotide substitutions per site”, however, in time-calibrated trees such as those produced by BEAST2, branch lengths are scaled by time, not by substitutions.

- Apart from the presented Bayesian tree, substitution trees should also be constructed to assess genome placement and branch lengths compared to previously published genetic diversity.

- In that regard, the estimation of genetic diversity among the analysed genomes could give clues on Y. pestis evolution within local plague reservoirs. How many SNP differences are identified among the newly generated genomes?

- Within the phylogenetic analysis, an indication of isolation hosts should be incorporated in all presented trees, as this is one of the correlation metrics dealt with in the present study.

- Line 296: “Effective sample sizes for the plasmid trees were high…Posterior probability score ranges were low: pMT1 [0.15, 0.48], pCD1 [0.10, 0.44], and pPCP1 [0.01, 0.11].” It is unclear how these metrics were calculated and what is their meaning. Moreover, their significance is unclear and should be clarified further.

Validity of the findings

Discussion:

- Line 376: “This finding suggests that the protein can change somewhat in the flea host, but that it must undergo negative selection to maintain function in the murine host.” This conclusion is unclear and should be clarified further.

- A previous study by Cui et al., 2020, provided indications for selection pressures on a difference phylogenetic branch (0.ANT, isolated from the Xinjiang Uygur Autonomous Region) involving the gene rpoZ, potentially associated with biofilm production. How are these results interpreted in light of the findings presented here?

Additional comments

Additional:

- I was not able to find a data availability statement within the paper. The authors should make the produced raw data publicly available prior to publication of the paper. Relevant accession IDs should be made available during the review phase of the paper.

---

## Round 0.2 · accepted · Accept

We appreciate your prompt response to the reviewers' concerns. Your contribution is interesting and timely. Thank you for considering PeerJ as an appropriate outlet for this research.